# Clinical Characteristics of Obstructive Sleep Apnea in Psychiatric Disease

**DOI:** 10.3390/jcm8040534

**Published:** 2019-04-18

**Authors:** Beat Knechtle, Nicholas-Tiberio Economou, Pantelis T. Nikolaidis, Lemonia Velentza, Anastasios Kallianos, Paschalis Steiropoulos, Dimitrios Koutsompolis, Thomas Rosemann, Georgia Trakada

**Affiliations:** 1Institute of Primary Care, University of Zurich, 8091 Zurich, Switzerland; beat.knechtle@hispeed.ch (B.K.); thomas.rosemann@usz.ch (T.R.); 2Division of Pulmonology, Department of Clinical Therapeutics, National and Kapodistrian University of Athens School of Medicine, Alexandra Hospital, 11528 Athens, Greece; nt_economou@yahoo.it (N.-T.E.); lemvele@gmail.com (L.V.); kallianos.pulm@gmail.com (A.K.); dkg348483@gmail.com (D.K.); 3Exercise Physiology Laboratory, 18450 Nikaia, Greece; pademil@hotmail.com; 4Department of Pulmonology, Democritus University of Thrace Medical School, University Hospital of Alexandroupolis, 68100 Alexandroupolis, Greece; steiropoulos@yahoo.com

**Keywords:** obstructive sleep apnea, psychiatric disease, major depressive disorder, bipolar disorder, schizophrenia and psychotic disorder, sleep study

## Abstract

Patients with serious psychiatric diseases (major depressive disorder (MDD), bipolar disorder (BD), and schizophrenia and psychotic disorder) often complain about sleepiness during the day, fatigue, low energy, concentration problems, and insomnia; unfortunately, many of these symptoms are also frequent in patients with Obstructive Sleep Apnea (OSA). However, existing data about the clinical appearance of OSA in Psychiatric Disease are generally missing. The aim of our study was a detailed and focused evaluation of OSA in Psychiatric Disease, in terms of symptoms, comorbidities, clinical characteristics, daytime respiratory function, and overnight polysomnography data. We examined 110 patients (56 males and 54 females) with stable Psychiatric Disease (Group A: 66 with MDD, Group B: 34 with BD, and Group C: 10 with schizophrenia). At baseline, each patient answered the STOP–Bang Questionnaire, Epworth Sleepiness Scale (ESS), Fatigue Severity Scale (FSS), and Hospital Anxiety and Depression Scale (HADS) and underwent clinical examination, oximetry, spirometry, and overnight polysomnography. Body Mass Index (BMI), neck, waist, and hip circumferences, and arterial blood pressure values were also measured. The mean age of the whole population was 55.1 ± 10.6 years. The three groups had no statistically significant difference in age, BMI, hip circumference, and systolic and diastolic arterial blood pressure. Class II and III obesity with BMI > 35 kg/m^2^ was observed in 36 subjects (32.14%). A moderate main effect of psychiatric disease was observed in neck (*p* = 0.044, *η*^2^ = 0.064) and waist circumference (*p* = 0.021, *η*^2^ = 0.078), with the depression group showing the lowest values, and in pulmonary function (Forced Vital Capacity (FVC, %), *p* = 0.013, *η*^2^ = 0.084), with the psychotic group showing the lowest values. Intermediate to high risk of OSA was present in 87.37% of participants, according to the STOP–Bang Questionnaire (≥3 positive answers), and 70.87% responded positively for feeling tired or sleepy during the day. An Apnea–Hypopnea Index (AHI) ≥ 15 events per hour of sleep was recorded in 72.48% of our patients. AHI was associated positively with male sex, schizophrenia, neck, and waist circumferences, STOP–Bang and ESS scores, and negatively with respiratory function. A large main effect of psychiatric medications was observed in waist circumference (*p* = 0.046, *η*^2^ = 0.151), FVC (%) (*p* = 0.027, *η*^2^ = 0.165), and in time spend with SaO_2_ < 90% (*p* = 0.006, *η*^2^ = 0.211). Our study yielded that patients with Psychiatric Disease are at risk of OSA, especially men suffering from schizophrenia and psychotic disorders that complain about sleepiness and have central obesity and disturbed respiratory function. Screening for OSA is mandatory in this medical population, as psychiatric patients have significantly poorer physical health than the general population and the coexistence of the two diseases can further negatively impact several health outcomes.

## 1. Introduction

Obstructive sleep apnea (OSA) is a chronic condition characterized by repetitive collapse of the upper airway during sleep, leading to oxygen desaturation, sympathetic activation, and recurrent arousals. It is associated with considerable morbidity and mortality and numerous studies indicate a causal relationship between OSA and hypertension, cardiovascular disease, insulin resistance, diabetes mellitus, and neurocognitive dysfunction [1,2,3]. OSA is diagnosed when (1) polysomnography (PSG) or home sleep apnea testing (HSAT) demonstrates ≥ 5 obstructive respiratory events per hour of sleep and one or more of the following symptoms are present: (a) The patient complains of sleepiness, non-restorative sleep, fatigue, or insomnia symptoms; (b) the patient wakes with breath-holding, gasping, or choking; (c) the bed partner or other observer reports habitual snoring, breathing interruptions, or both during the patient’s sleep; or (d) the patient has been diagnosed with hypertension, mood disorder, cognitive dysfunction, coronary artery disease, stroke, congestive heart failure, atrial fibrillation, or type 2 diabetes mellitus; or (2) PSG or HSAT demonstrates ≥ 15 obstructive respiratory events per hour of sleep, independently of symptoms and comorbidities [4].

OSA is a prevalent disorder among general population. Recently, in Northern Europe (Switzerland), the HypnoLaus study (2009–2013) recorded a prevalence of moderate-to-severe Sleep Disordered Breathing (SDB) (≥15 events per hour) 23.4% in women and 49.7% in men [5]. The reported prevalence of OSA has increased over time due to the ongoing epidemic of obesity and due to more sophisticated polysomnographic recording techniques and revised diagnostic criteria [5,6]. Although some of the symptoms of OSA are associated with psychiatric disease (Psychiatric Disease), few studies examine the prevalence and the clinical characteristics of OSA in individuals with diagnosed Psychiatric Disease [7]. Several psychiatric patients report insomnia and complain of poor quality of sleep, reduction in total sleep time, and frequent nighttime awakenings. Others report hypersomnolence and complain of prolonged night time sleep, difficulty in wakening, and excessive daytime sleepiness. Tiredness, lack of energy, concentration problems, sleepiness or hypersomnolence, and insomnia are common complaints of both OSA and Psychiatric Disease [7].

Furthermore, it is well documented that visceral obesity—excessive deposition of fat in the abdominal viscera and omentum—is associated with OSA [3]. Apnea-Hypopnea Index (AHI) significantly correlates with intra-abdominal fat and waist circumference is a better predictor of OSA than Body Mass Index (BMI) [8]. The prevalence of obesity is greater in psychiatric population than in general population [9]. Chronically stressed individuals frequently have mild hypercortisolism, due to a blunted circadian rhythm of cortisol secretion and stress-induced, glucocorticoid-mediated visceral obesity and metabolic syndrome manifestations [10]. Moreover, specifically in patients with schizophrenia, the presence of metabolic syndrome is also associated with an increased prevalence of a restrictive pattern of breathing, indicative of reduced lung volumes [11]. Psychiatric Disease is also associated with an increased risk of respiratory diseases [12,13]. In a population–based study, people with psychosis had significantly lower lung function values compared with the general population, even after correcting for smoking and other potential confounders [13]. Impaired daytime respiratory function may further aggravate instability of breathing during sleep, leading to severe hypoventilation and significant hypoxemia.

As OSA and Psychiatric Disease both share similar symptoms—poor quality of sleep or prolonged sleep—and possible underlying mechanisms—visceral obesity or impaired lung function—one can hypothesize that OSA is a common disease among psychiatric patients. Indeed, in a recent review, it appears that there is an elevated prevalence of the disease in psychiatric patients with major depressive disorder (MDD) and post-traumatic stress disorder (PTSD) [7]. Another meta-analysis also confirms that 25.7% of patients with Psychiatric Disease have OSA, more frequently those with MDD (36.3%) when compared to bipolar disortder (BD) (24.5%) and schizophrenia (15.4%) [14]. However, findings are still unclear due to the small number of studies and the heterogeneity of methods and diagnostic criteria. We conducted this study in order to better clarify the clinical picture of OSA in Psychiatric Disease, in relation to metabolic indices and daytime respiratory function of these patients and to evaluate the impact of medications.

## 2. Experimental Section

### 2.1. Study Population

We examined 110 outpatients (56 males and 54 females) with a diagnosis of Psychiatric Disease, according to standardized diagnostic criteria (e.g., DSM-V (American Psychiatric Association, 2013) or ICD-10 [15] (Group A: 66 with MDD (F33), Group B: 34 with BD (F31), and Group C: 10 with schizophrenia and psychotic disorder (F20)). The study was conducted between 1 January 2016 and 30 June 2018, in the Division of Pulmonology, at the Department of Clinical Therapeutics of the National and Kapodistrian University of Athens School of Medicine, at Alexandra Hospital of Athens. The patients were referred to us from the 1st Department of Psychiatry of Eginition University Hospital, and the assessment was conducted by one pulmonologist (Dr. G.T.). The ethical committee of Alexandra Hospital approved the study protocol and all subjects gave consent to participate in the study after appropriate information was given.

### 2.2. Clinical Assessment and Spirometry Measurements

At baseline, each patient underwent detailed medical history—including the number, the type, and the medications of other coexisting diseases—and clinical examination. BMI, neck, waist, and hip circumferences, and arterial blood pressure values were also measured. Respiratory function was evaluated with oximetry (model 8800, Nonin Medical, Inc., Plymouth, MN, USA), and spirometry (Master screen Diffusion, Jaeger, Germany). Spirometry was performed by a specially trained technician, with the patient in a sitting position, according to the manufacturer’s instructions and standard European Respiratory Society (ERS)/American Thoracic Society (ATS) guidelines [16]. The main outcome variables were Forced Vital Capacity (FVC) (i.e., the maximal volume of air that can be forcefully expelled from the lungs after maximal inhalation, in liters) and Forced Expiratory Volume in 1 s (FEV_1_) (i.e., the volume of air that can be forcefully expelled from the lungs in the first second after maximal inhalation, in liters). The patient’s age, height, and weight were recorded for use in the calculation of reference values.

### 2.3. Questionnaires

All patients answered the snoring, tiredness, observed apnea, high blood pressure, BMI, age, neck circumference, and male gender (STOP–Bang) questionnaire, Epworth Sleepiness Scale (ESS), Fatigue Severity Scale (FSS), and Hospital Anxiety and Depression Scale (HADS). The STOP–Bang questionnaire is a concise and easy to-use, self-reportable screening tool for OSA that includes eight yes or no questions about snoring, tiredness, observed apnea, high blood pressure, BMI > 35 kg/m^2^, age > 50 years old, neck circumference > 40 cm, and male gender [17]. Low risk for OSA is defined as ≤ 2 positive answers, intermediate risk as 3–4, and high risk ≥ 5. The sensitivity of a STOP–Bang score ≥ 3 to detect moderate to severe OSA (AHI > 15) and severe OSA (AHI > 30) is 93% and 100%, respectively [17].

The ESS is a self-report questionnaire that evaluates the tendency to fall asleep in eight daily situations. The ESS score ranges from 0 to 24, and a score ≥ 10 indicates excessive daytime sleepiness [18]. FSS is a questionnaire with nine questions estimating the fatigue severity in different situations during the past week. Grading ranges from 1 (strong disagreement) to 7 (strong agreement), where the final score is the mean value of the nine items, and a score ≥ 4 is interpreted as fatigue [19]. HADS is also a self-report, 14-item, depression and anxiety screening instrument assessing the severity of symptoms in medically ill patients, while excluding physical symptoms potentially attributable to comorbid medical conditions [20]. Each question is scored from 0–3, with a total score up to 21 for each subscale, while scores >10 are suggestive either for depressive (HAD-D) or anxiety (HAD-A) symptoms.

### 2.4. Sleep Study

A standard full night polysomnography (PSG) was performed on each patient on the same night of the initial evaluation (Alice, Respironics, Murraysville, PA, USA). Sleep records were scored according to standard criteria and manually revised by an expert [21]. OSA was diagnosed as an AHI of ≥ 5 events per hour of sleep with associated symptoms or comorbidities or an AHI of ≥ 15, regardless of associated symptoms or comorbidities, according to the American Academy of Sleep Medicine (AASM) new diagnostic criteria [4].

### 2.5. Data Analysis

Possible associations between metabolic indices, respiratory function, and sleep data were explored. Number and type of comorbidities were compared between only Psychiatric Disease and OSA plus Psychiatric Disease patients. In addition, patients were also classified into seven groups, depending on the psychiatric medications they received, according to the Anatomical Therapeutic Chemical (ATC) classification system [22], i.e., N03 (antiepileptics) (*n* = 2), N05 (psycholeptics) (*n* = 6), N06 (psychoanaleptics) (*n* = 29), combination of N03, and N05 (*n* = 5), or N03, or N06 (*n* = 7), N05, and N06 (*n* = 21), or N03, N05, N06 (*n* = 15), and were compared for BMI, neck, waist, and hip circumferences, FEV1 (%), FVC (%), ESS, FSS, HADS-A, HADS-D, AHI, meanSaO_2_ (%), minSaO_2_ (%), and time spend with SaO_2_ < 90%.

### 2.6. Statistical Analysis

Statistical analysis included summarization of the data in tables and charts and was performed by using an IBM SPSS Statistics version 25 program. The values of parameters were expressed as mean and standard deviation (M ± SD). The Kolmogorov–Smirnov test and visual inspection of normal Q-Q plots were used to examine normality. One-way analysis of variance (ANOVA) was conducted to compare age, body mass index, lung function tests, and PSG parameters and one-way analysis of covariance (ANCOVA) was conducted to compare the questionnaires’ scores (ESS, FSS, HADS-A, and HADS-D) over the three groups. Univariate and multivariate analyses were used to determine possible correlations between anthropometric data and questionnaires’ scores and AHI. The magnitude of the differences among Psychiatric Disease groups was evaluated by eta squared (*η*^2^), which was interpreted as small (0.01 ≤ *η*^2^ < 0.06), medium (0.06 ≤ *η*^2^ < 0.14), and large (*η*^2^ ≥ 0.14), respectively, as eta squared has been widely used as an estimate of effect size when applying ANOVA [23]. A *p* value lower than 0.05, was considered as being significant.

## 3. Results

### 3.1. Demographic Data

All patients were referred to us with a clinical suspicion of sleep apnea mainly because of unrefreshing sleep and excessive daytime sleepiness and/or fatigue. The mean age of the total population was 55.1 ± 10.6 years, with a mean BMI 32.6 ± 6.6 kg/m^2^. All patients were under appropriate therapy with either psychoanaleptics (N06) (that is psychostimulants, antidepressants, and combination of antidepressants with anxiolytics), or psycholeptics (N05) (that is antipsychotics, anxiolytics, and sedatives/hypnotics), or antiepileptics (N03), alone or in combination. The mean self-reported hours of sleep per day were 8.3 ± 2.6 (range: 3.5–17), with no statistically significant difference between the three groups (A: 8.0 ± 2.3, B: 8.7 ± 3.0, and C: 9.7 ± 3.0). More than 9 h of sleep was reported by 34.61% of patients, 7–9 h by 39.74%, and less than 7 h by 25.64%.

### 3.2. Respiratory and Metabolic Indices

The baseline characteristics and daytime respiratory function of the study population are presented in Table 1. A moderate main effect of psychiatric disease was observed in neck (*p* = 0.044, *η*^2^ = 0.064) and waist circumference (*p* = 0.021, *η*^2^ = 0.078), with the depression group showing the lowest values, and in pulmonary function (FVC, *p* = 0.013, *η*^2^ = 0.084), with the psychotic group showing the lowest values.

### 3.3. Questionnaires

In the whole population, 12.62% of participants answered positively to 0–2 questions in the STOP–Bang Sleep Apnea questionnaire, whereas 87.37% answered positively to > 2 (3–8) questions. According to the total score, 12.62% subjects were at low risk of OSA, 38.83% at intermediate risk, and 48.54% at high risk of OSA. The distribution of the self-reported symptoms and the demographic items are summarized in the Appendix A. The ESS score was >10 in 30%, whereas the FSS score was ≥ 4 in 75% of patients. In the univariate analysis, the score in FSS—but not in ESS—correlated with both HADS-A and -D scores (*r* = 0.460, *p* = 0.012, and *r* = 0.490, *p* = 0.007, respectively). The results in different questionnaires are presented in Table 2. Also, the self-reported sleep duration did not correlate with ESS (*r* = 0.17, *p* = 0.170), FSS (*r* = −0.10, *p* = 0.657), HADS-A (*r* = −0.08, *p* = 0.551), and HADS-D (*r* = 0.014, *p* = 0.911).

### 3.4. Sleep Data

An AHI ≥15 events per hour of sleep was recorded in 72.48% of our patients, whereas an AHI ≥ 5 plus symptoms or comorbidities was recorded in 84.4%. Sleep characteristics are presented in Table 3. Psychotic patients slept more in N1 stage and had a higher AHI and more severe hypoxemia during sleep (mean and min SaO_2_, time spent in SaO_2_ < 90%) when compared to depressive and bipolar patients. In the multivariate analysis, AHI correlated positively with male gender (*p* < 0.001), neck and waist circumferences (*p* < 0.001, and *p* = 0.030, respectively), and STOP–Bang and ESS scores (*p* = 0.004, and *p* = 0.009, respectively), and negatively with respiratory function (FEV_1_, *p* = 0.037, and FVC, *p* = 0.003). The sensitivity of a STOP–Bang score ≥ 3 to detect moderate and severe OSA (AHI ≥ 15) and mild OSA (AHI ≥ 5) plus symptoms or comorbidities was 92% and 88.51%, respectively. Corresponding negative predictive values were 45.45% and 16.66%.

Minimum and mean SaO_2_ correlated positively with respiratory function (FVC, *p* < 0.001, and *p* = 0.006, and FEV_1_, *p* = 0.006, and *p* = 0.019, respectively) and negatively with neck (*p* = 0.005, and, *p* < 0.001, respectively) and waist (*p* = 0.008, and *p* = 0.002, respectively) circumferences. Mean SaO_2_ correlated also negatively with BMI (*p* = 0.023). Finally, time spent on SaO_2_ < 90% correlated negatively with respiratory function (FVC, *p* < 0.001, and FEV_1_, *p* < 0.001) and positively with neck circumference (*p* = 0.047).

### 3.5. Comorbidities

Only seventeen (17) psychiatric patients without OSA (64.71%) and 13 psychiatric patients with coexisting OSA (14.13%) had no other known diseases (*p* < 0.001). All others suffered from 1 to ≥ 3 comorbidities. The most common comorbidities were hypertension, dyslipidemia, diabetes mellitus (DM), and cardiovascular disease. Obesity was very prevalent in all this population. Only one to four (23.53%) psychiatric patients were without OSA and one to ten (9.78%) psychiatric patients with OSA had a normal BMI. Class I and II obesity (BMI > 35 kg/m^2^) was present in 32.14% of all participants.

### 3.6. Medications

A large main effect of psychiatric medications was observed in waist circumference (*p* = 0.046, *η*^2^ = 0.151), FVC (%) (*p* = 0.027, *η*^2^ = 0.165), and time spent with SaO_2_ < 90% (*p* = 0.006, *η*^2^ = 0.211) (Table 4). It should be highlighted that in the insignificant findings of the comparison among medication groups, a moderate-to-large effect size was shown.

## 4. Discussion

In our sample of outpatients with serious mental illness (major depression, bipolar disease, or schizophrenia), we demonstrated a very high prevalence of sleep apnea. Comorbidities—and especially cardio–metabolic—were more prevalent in patients with coexisting Psychiatric Disease and OSA when compared to Psychiatric Disease-only patients. An impact of medications was observed in waist circumference, FVC (%), and time spent with SaO_2_ < 90% during sleep. The typical patient was a man, suffering from schizophrenia and complaining about sleepiness, with central obesity, as expressed by waist circumference, and disturbed daytime respiratory function, as expressed by FVC.

OSA is a very prevalent disease in general unselected populations, based on PSG data, ranging from 46.6 to 49.7% in men and 23.4 to 30.5% in women [5,24]. Although it is a relatively common sleep disorder, up to 80% of individuals with moderate to severe OSA may remain undiagnosed and, more alarmingly, untreated [25], and this percentage can be even greater in the particular subpopulation of psychiatric patients. We previously described a patient with bipolar disorder who was diagnosed with OSA during a resistant, depressive episode and switched to mania, after 10 days of treatment with continuous positive airways pressure (CPAP) during sleep [26]. We also described two patients with bipolar disorder who presented complications during anesthesia for electroconvulsive therapy, possibly due to underlying, undiagnosed OSA [27]. These cases indicate that failure to diagnose and treat sleep apnea in psychiatric patients may worsen the severity of their psychiatric disease and can expose them to several complications. However, OSA in the psychiatric population has not been yet thoroughly evaluated and usually remains under-diagnosed.

Standardized questionnaires for the identification of OSA, such as the widely used STOP–Bang questionnaire and ESS, are not validated on psychiatric patients. A previous study using the STOP–Bang screening tool found that 69% of the psychiatric patients were positive for at least three items in STOP–Bang questionnaire, and 20% had a positive ESS [28]. In our population, 48.54% of participants were at high and 38.83% at intermediate risk for OSA. Although 68.87% responded affirmatively about “feeling tired, or fatigued, or sleepy during daytime”, only 30% had an ESS score > 10, whereas the FSS score was ≥ 4 in 75% of patients. Sleepiness was associated with AHI, whereas fatigue was associated with symptoms of anxiety and depression.

People with Psychiatric Disease, particularly MDD, had a high prevalence of OSA, ranging from 13.9 to 42.4% in different clinical studies [7,29]. Higher frequencies of OSA were seen in MDD (36.3%) than in BD (24.5%) and schizophrenia (15.4%) [29]. Our study revealed a higher prevalence of sleep apnea among psychiatric patients when compared to previous studies, but it was the first according to AASM new diagnostic criteria [4]. Current evidence emphasizes the importance of using arousal-based scoring of respiratory events when evaluating patients suspected of having OSA, as not utilizing arousal-based scoring may lead to a missed diagnosis in up to 30% to 40% of patients with OSA [30]. Using different criteria for scoring hypopneas can lead to marked differences in AHI.

Psychiatric disorders have independent associations with obesity, metabolic dysfunction, and cardiovascular disease, which are all independent predictors of OSA [7,9]. Moreover, previous researchers hypothesized that OSA, psychological symptoms, and metabolic disease are the diverse consequences of underlying biological, metabolic, and neurologic dysregulation, through low-grade inflammation, oxidative and nitrosative stress, and neurotransmitter imbalances [31,32]. Notably, Gupta and Simpson [7] proposed a feed-forward pathway for the development of symptoms of upper airway instability and OSA from biological, psychiatric, and metabolic dysregulation. Each symptom was an independent entry point to the cycle. If left untreated, the likelihood of the synergistic development of more symptoms increases, resulting in OSA.

A great percentage of our patients were obese. In accordance with previous studies, waist circumference, and not BMI, of our population correlated with AHI [31]. Measurement of abdominal obesity is strongly associated with increased cardio–metabolic risk, cardiovascular events, and mortality. Although waist circumference is a crude measurement, it correlates with obesity and visceral fat amount, and is a surrogate marker for insulin resistance [33]. Specifically in patients with schizophrenia, metabolic dysfunction is also associated with an increased prevalence of restrictive lung pattern of breathing [11]. Our findings were similar; the three groups did not differ in terms of BMI, however, patients with schizophrenia had higher waist circumferences and lower FVCs, and therefore, higher AHI and more severe hypoxemia during sleep when compared to MDD and BD patients. Moreover, the coexistence of OSA and Psychiatric Disease leads to significantly more cardio–metabolic comorbidities than Psychiatric Disease alone.

Another major issue that was pointed out from this study was the effect of psychiatric medications on metabolic and respiratory dysfunction of our patients. It is well known that psychiatric medications result in weight gain and obesity [34,35,36]. Also, benzodiazepines may have direct effects on breathing during sleep which result in airway obstruction [37], whereas antipsychotics are associated with an increased AHI, independently of BMI and neck circumference [36]. It seems that administration of these drugs further aggravates metabolic syndrome and airway resistance, which in turn stimulate underlying pathophysiological mechanisms leading to OSA.

A limitation of our study was that psychiatric patients were referred to us with a clinical suspicion of sleep-disordered breathing and were not randomly selected from the general population. A sample with clinical signs of OSA is significantly more predisposed to a high rate of OSA than a random sample of individuals with psychiatric diseases [7]. Also, patients with schizophrenia were only males. Moreover, the total size of our population was small. Finally, as these patients performed a unique polysomnographic study, the “first-night effect” may have had an impact on the sleep recordings [38]. On the other hand, the strength of the study was that we described in detail sleep apnea in psychiatric patients, in terms of symptoms, metabolic indices, daytime respiratory function, and comorbidities. Popular screening questionnaires, like STOP–Bang and ESS, were also evaluated in this particular subpopulation of patients. As a recent study has stated, symptoms of sleep disorders are very common in psychiatric patients and should be further confirmed by using appropriate questionnaires and diagnostic tests [39]. Finally, we pointed out the impact of psychiatric medications in metabolic indices, as expressed by waist circumference, and in daytime and nighttime respiratory function, as expressed by FVC and time spent with SaO_2_ <90% during sleep.

## 5. Conclusions

This study further highlights the importance of examining this particular subpopulation of patients for risk factors and symptoms associated with sleep apnea. According to our findings, OSA is a very prevalent disease among psychiatric patients. Although the symptoms of OSA can be also attributed to psychiatric disease or to side effects of psychotropic medications, STOP–Bang is able to identify most patients with OSA, especially when ESS is also positive. STOP–Bang and ESS scores, male gender, neck and waist circumferences, and daytime respiratory function correlate with the severity of apnea, in terms of AHI and hypoxemia during sleep. Medications may elicit some of the underlying pathophysiological mechanisms of OSA. Finally, coexistence of OSA and Psychiatric Disease leads to significant cardio–metabolic comorbidity and probably contributes to the poor physical health of the psychiatric patients. The impact of psychiatric medications on OSA in Psychiatric Disease should be carefully evaluated on future, larger populations.

## Figures and Tables

**Table 1 jcm-08-00534-t001:** Baseline characteristics and daytime respiratory function of the study groups (mean ± SD).

	Total Population (*n* = 110)	Major Depressive Disorder (MDD) (*n* = 66)	Bipolar Disorder (BD) (*n* = 34)	Schizophrenia and Psychotic Disorder (*n* = 10)
Sex (men/women)	54/56	29/37	15/19	10/0 ^†^
Age (years)	55.1 ± 10.6	57.2 ± 10.4	52.6 ± 10.6	51.0 ± 9.8
BMI (kg/m^2^)	32.6 ± 6.6	32.1 ± 7.1	34.0 ± 6.1	31.3 ± 4.3
Neck circumference (cm)	40.9 ± 5.2	40.2 ± 5.3	41.1 ± 4.3	44.8 ± 5.9 *
Waist circumference (cm)	113.0 ± 14.3	109.9 ± 15.4	117.0 ± 11.3	120.7 ± 9.1 *
Hip circumference (cm)	112.3 ± 12.0	110.8 ± 12.2	114.9 ± 12.5	113.8 ± 5.8
Systolic Arterial Blood Pressure (mmHg)	122.6 ± 16.2	122.2 ± 15.2	123.5 ± 18.7	122.5 ± 15.6
Diastolic Arterial Blood Pressure (mmHg)	79.4 ± 10.3	78.6 ± 10.3	79.4 ± 11.1	84.4 ± 7.3
FVC (L)	3.21 ± 1.01	3.20 ± 1.05	3.20 ± 0.99	3.36 ± 0.91
FVC (%)	91.4 ± 16.8	94.1 ± 17.1	90.0 ± 14.5	77.0 ± 15.7 *
FEV_1_ (L)	2.72 ± 1.09	2.64 ± 0.95	2.86 ± 1.39	2.80 ± 0.76
FEV_1_ (%)	91.9 ± 17.4	94.3 ± 18.5	90.7 ± 13.4	79.6 ± 17.8
FEV_1_/FVC	82.8 ± 6.2	82.1 ± 6.6	84.0 ± 5.7	83.5 ± 5.1
PEF (L/min)	6.40 ± 2.31	6.48 ± 2.24	6.24 ± 2.38	6.36 ± 2.83
PEF (%)	88.2 ± 23.5	91.5 ± 21.7	85.4 ± 23.0	74.5 ± 32.9
SaO_2_ (%)	96.1 ± 1.8	96.3 ± 1.7	95.8 ± 2.0	96.2 ± 2.3

* *p* < 0.05, MDD vs. schizophrenia and other psychotic diseases; ^†^
*p* < 0.05, difference in the distribution of sex by group. BMI: Body Mass Index, FVC: Forced Vital Capacity, FEV_1_: Forced Expiratory Volume in 1sec, PEF: Peak Expiratory Flow Rate, SaO_2_: Oxyhemoglobin Saturation.

**Table 2 jcm-08-00534-t002:** Epworth Sleepiness Scale (ESS), Fatigue Severity Scale (FSS), and Hospital Anxiety and Depression Scale (HADS) of the study groups (mean ± SD).

	Total Population (*n* = 110)	Major Depressive Disorder (MDD) (*n* = 66)	Bipolar Disorder (BD) (*n* = 34)	Schizophrenia and Psychotic Disorder (*n* = 10)
ESS	7.9 ± 5.0	8.5 ± 5.0	6.8 ± 4.9	8.2 ± 5.2
FSS	5.7 ± 1.9	5.2 ± 2.2	6.4 ± 0.8	7.0 ± 0.0
HADS A	10.3 ± 5.6	9.2 ± 5.0	12.2 ± 5.8	10.1 ± 6.9
HADS D	10.4 ± 5.6	9.0 ± 5.0	12.6 ± 5.8 *	10.5 ± 5.9

* *p* < 0.05, MDD vs. BD

**Table 3 jcm-08-00534-t003:** Sleep characteristics of the study groups (mean ± SD).

	Total Population (*n* = 110)	Major Depressive Disorder (MDD) (*n* = 66)	Bipolar Disorder (BD) (*n* = 34)	Schizophrenia and Psychotic Disorder (*n* = 10)	*P* Value	*η*^2^ (Effect Size)
TRT (min)	348.2 ± 99.6	352.3 ± 99.3	355.7 ± 88.0	291.5 ± 132.3	0.202	0.030
TST (min)	273.5 ± 98.3	275.9 ± 103.4	280.9 ± 89.6	228.8 ± 87.5	0.355	0.020
SE (%)	79.0 ± 17.0	77.7 ± 19.1	80.8 ± 13.0	81.2 ± 12.7	0.641	0.008
Sleep onset (min)	32.3 ± 37.5	30.8 ± 30.5	34.4 ± 36.7	34.6 ± 75.6	0.888	0.002
WASO (min)	50.0 ± 65.5	48.6 ± 70.5	58.6 ± 63.2	19.8 ± 27.4	0.751	0.020
N1 (min)	23.8 ± 33.1	24.2 ± 34.5	19.9 ± 27.5	34.7 ± 42.4	0.487	0.014
N1 (%)	10.5 ± 15.6	10.0 ± 13.1	7.9 ± 10.7	22.8 ± 34.8	0.037	0.061
N2 (min)	183.7 ± 90.5	180.7 ± 93.3	195.4 ± 77.6	161.8 ± 116.3	0.567	0.011
N2 (%)	67.0 ± 22.9	66.0 ± 22.8	69.8 ± 19.1	64.2 ± 35.8	0.688	0.007
N3 (min)	39.5 ± 43.7	38.9 ± 43.3	44.0 ± 44.1	27.2 ± 48.2	0.587	0.010
N3 (%)	14.0 ± 15.4	13.9 ± 15.6	14.9 ± 14.1	11.0 ± 19.1	0.795	0.004
REM (min)	24.1 ± 31.6	28.4 ± 36.4	20.7 ± 21.1	5.1 ± 13.2	0.087	0.046
REM (%)	8.9 ± 12.3	10.5 ± 14.3	7.4 ± 7.9	2.1 ± 5.4	0.112	0.041
Apneas (*n*)	151.7 ± 134.1	142.3 ± 122.6	147.0 ± 143.4	237.9 ± 164.1	0.130	0.038
Obstructive (*n*)	49.6 ± 85.3	41.9 ± 63.2	44.9 ± 82.3	118.6 ± 172.4	0.038	0.064
Central (*n*)	19.4 ± 44.8	21.4 ± 49.0	14.2 ± 32.4	24.4 ± 55.9	0.721	0.007
Mixed (*n*)	6.9 ± 17.3	8.5 ± 20.8	3.4 ± 8.0	9.0 ± 15.9	0.390	0.019
Hypopneas (*n*)	83.1 ± 77.7	79.2 ± 73.6	89.9 ± 90.7	85.9 ± 58.5	0.815	0.004
Mean Duration (s)	19.6 ± 4.8	19.8 ± 4.5	18.7 ± 4.3	22.2 ± 7.8	0.161	0.035
Max Duration (s)	49.8 ± 26.0	48.8 ± 22.9	46.8 ± 23.0	67.9 ± 47.1	0.085	0.047
AHI	35.3 ± 27.3	33.3 ± 24.8	30.9 ± 25.9	66.1 ± 33.6 *	0.001	0.119
AHI (nREM)	34.0 ± 28.4	31.9 ± 26. 1	28.9 ± 26.0	66.5 ± 33.4 *	0.001	0.131
AHI (REM)	18.3 ± 25.7	19.2 ± 27.1	20.0 ± 25.3	6.7 ± 13.2	0.364	0.021
minSaO_2_ (%)	82.9 ± 7.7	84.2 ± 6.1	82.2 ± 8.2	75.3 ± 12.4	0.006	0.096
meanSaO_2_ (%)	92.1 ± 2.8	92.5 ± 2.6	92.2 ± 2.3	88.4 ± 4.2 *	<0.001	0.142
SaO_2_ < 90% (min)	23.8 ± 44.5	20.9 ± 41.5	20.5 ± 33.2	61.2 ± 85.0	0.046	0.058
SaO_2_ < 90% (%)	16.7 ± 26.3	13.8 ± 23.9	15.4 ± 25.8	45.2 ± 33.1	0.005	0.098

TRT: Total Recording Time, TST: Total Sleep Time, SE: Sleep Efficiency, WASO: Wake after Sleep Onset, N1: Stage 1, N2: Stage 2, N3: Stage 3–4, REM: Rapid Eye Movement, AHI: Apnea-Hypopnea Index, SaO2: Oxyhemoglobin Saturation. Considering the existence of multiple comparisons, the Bonferroni correction (i.e., alpha level/number of comparisons, i.e., 0.05/27) was applied setting the significance level, respectively, to *p* = 0.002. * different from MDD and BD.

**Table 4 jcm-08-00534-t004:** Impact of psychotropic medications on BMI, neck, waist, and hip circumferences, FEV1 (%), FVC (%), AHI, meanSaO_2_ (%), minSaO_2_ (%), and time spend with SaO_2_ < 90% during sleep.

Variable	Medication Group							*p* Value	*η*^2^ (Effect Size)
	N03 (*n* = 2)	N05 (*n* = 6)	N06 (*n* = 29)	N03, N05 (*n* = 5)	N03, N06 (*n* = 7)	N05, N06 (*n* = 21)	N03, N05, N06 (*n* = 15)		
BMI (kg/m^2^)	28.6 ± 2.9	29.8 ± 6.0	30.2 ± 7.1	32.3 ± 5.8	32.8 ± 3.5	34.4 ± 5.5	35.7 ± 6.3	0.078	0.132
Neck circumference (cm)	42.0 ± 8.5	43.3 ± 7.3	38.5 ± 4.3	40.2 ± 4.8	42.9 ± 5.9	42.0 ± 4.0	41.1 ± 5.2	0.128	0.120
Waist circumference (cm)	109.5 ± 3.5	113.3 ± 12.3	106.1 ± 16.0	115.2 ± 16.8	117.1 ± 13.2	117.0 ± 11.6	119.8 ± 10.6	0.046	0.151
Hip circumference (cm)	103.5 ± 7.8	110.8 ± 8.2	108.8 ± 11.4	112.8 ± 4.8	116.4 ± 9.3	116.4 ± 14.2	118.9 ± 11.6	0.091	0.131
FEV1 (%)	113.2 ± 26.2	82..4 ± 19.3	97.0 ± 12.7	86.2 ± 15.2	97.3 ± 15.6	90.4 ± 18.2	87.2 ± 16.1	0.091	0.129
FVC (%)	110.6 ± 5.7	81.1 ± 16.6	97.8 ± 14.3	83.1 ± 14.7	96.9 ± 15.7	89.1 ± 18.4	85.0 ± 14.9	0.027	0.165
AHI	31.8 ± 0.0	55.8 ± 32.0	31.3 ± 21.6	53.1 ± 50.9	50.1 ± 32.2	24.0 ± 18.9	32.1 ± 26.5	0.058	0.146
meanSaO_2_ (%)	95.0 ± 0.0	88.7 ± 4.4	92.6 ± 2.5	92.9 ± 2.4	92.6 ± 1.9	92.3 ± 3.0	91.3 ± 3.2	0.077	0.138
minSaO_2_ (%)	93.0 ± 0.0	76.0 ± 13.3	85.5 ± 5.7	83.2 ± 8.3	81.7 ± 7.3	83.3 ± 6.7	80.9 ± 10.7	0.131	0.120
time with SaO_2_ < 90% (min)	0.0 ± 0.0	58.3 ± 44.5	8.6 ± 14.6	14.6 ± 15.7	14.2 ± 12.8	18.3 ± 31.4	25.2 ± 32.8	0.006	0.211

AHI: Apnea-Hypopnea Index, BMI: Body Mass Index, FVC: Forced Vital Capacity, FEV_1_: Forced Expiratory Volume in 1s, SaO_2_: Oxyhemoglobin Saturation. Considering the existence of multiple comparisons, the Bonferroni correction (i.e., alpha level/number of comparisons, i.e., 0.05/10) was applied setting the significance level, respectively, to *p* = 0.005.

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
