# Peer review of "Clinical Characteristics of Obstructive Sleep Apnea in Psychiatric Disease"

_jcm, 2019, doi:10.3390/jcm8040534_

Round 1
Reviewer 1 Report
The addition of evaluating correlation between psychotropic medication and various metabolic, pulmonary and sleep measures is useful. The ATC classifications were new to me and it might help the reader to not have to refer to other literature if the classifications used were described with the more common clinical vernacular of antiepileptics, psycholeptics and psychoanaleptics or even more common: mood stabilizers, antipsychotics and antidepressants.
Author Response
Comments and Suggestions for Authors
The addition of evaluating correlation between psychotropic medication and various metabolic, pulmonary and sleep measures is useful. The ATC classifications were new to me and it might help the reader to not have to refer to other literature if the classifications used were described with the more common clinical vernacular of antiepileptics, psycholeptics and psychoanaleptics or even more common: mood stabilizers, antipsychotics and antidepressants.
We thank the reviewer for the useful comments. ATC classifications available by WHO are meant to summarize various drugs in certain categories; as a matter of fact we find it very useful to use ATC mode instead of referring lots of drugs, which, to our view, would have added rather confusion than clearness. Nevertheless, we added a small comment on the composition of these categories.
Reviewer 2 Report
Summary:
The paper is a detailed examination of OSA in a small sample of individuals with psychiatric disease. The authors conclude that OSA is common in those with psychiatric disease, particularly so in men who communicate their sleepiness and have schizophrenia/psychotic disorders.
Comments and questions:
I struggled a little with finding the novelty in this paper, for example, the Gupta review includes 49 papers examining OSA in psychiatric populations, with a total sample of nearly 2000 participants across papers, suggesting quite a large body of work in this area already. Perhaps the authors would consider making the novelty more apparent in the introduction, e.g., the intimate detail included in their examination of OSA in a psychiatric sample (e.g., medications) and/or comparison of subjective and objective OSA screen and utility in clinical settings. They cover these points well in the Discussion.
The authors have written a clear and concise paper, and have completed a good review of the relevant literature.
It may be a convention for these authors but I wonder if they would consider writing the full 'psychiatric disease' rather than the initialism, PD. Though they do define the initialism of 'PD' I continued to read 'Parkinson's Disease' rather than 'Psychiatric Disease', perhaps other readers will do the same.
Pg 5. Please include a note at the base of Table 1 explaining the cross after '0' for 'sex' x 'Schiz/other psych'.
Pg 5. 'of OSA' has a strikethrough, please edit.
Pg 5/6. I felt that Table 2 could be included as an Online Supplement and didn't need to be included in full in the paper.
I loved the comparison of STOP-BANG to PSG, makes screening very accessible in a clinic setting. I also thought it very thorough to divide medication by type and look at the effect on variables of interest.
Pg 8. Please restructure Table 5 so that it is easier to read, put the values and labels on one line. It gets hard to read when decimal points drop off to different lines.
The conclusions are well thought out and helpful for both clinicians and researchers.
Author Response
Comments and Suggestions for Authors
Summary:
The paper is a detailed examination of OSA in a small sample of individuals with psychiatric disease. The authors conclude that OSA is common in those with psychiatric disease, particularly so in men who communicate their sleepiness and have schizophrenia/psychotic disorders.
Comments and questions:
I struggled a little with finding the novelty in this paper, for example, the Gupta review includes 49 papers examining OSA in psychiatric populations, with a total sample of nearly 2000 participants across papers, suggesting quite a large body of work in this area already. Perhaps the authors would consider making the novelty more apparent in the introduction, e.g., the intimate detail included in their examination of OSA in a psychiatric sample (e.g., medications) and/or comparison of subjective and objective OSA screen and utility in clinical settings. They cover these points well in the Discussion.
The authors have written a clear and concise paper, and have completed a good review of the relevant literature.
The 2nd reviewer points out several important issues, and we appreciate a lot this contribute. Regarding the points stated:
Studies evaluating OSA in psychiatric patients with the use of proper psychiatric assessment, PSG and overall sophisticated investigation (like in our case) are scarce. As a matter of fact in Gupta’s paper, studies with these characteristics are very few; that is why we think our study may “add” useful information in the literature; moreover, this is the first study presenting data on OSA in psychiatric population from Greece.
It may be a convention for these authors but I wonder if they would consider writing the full 'psychiatric disease' rather than the initialism, PD. Though they do define the initialism of 'PD' I continued to read 'Parkinson's Disease' rather than 'Psychiatric Disease', perhaps other readers will do the same.
We perfectly agree that PD strictly relates to Parkinson’s Disease and not to Psychiatric Disease, thus we corrected as suggested.
Pg 5. Please include a note at the base of Table 1 explaining the cross after '0' for 'sex' x 'Schiz/other psych'.
We added an explanation.
Pg 5. 'of OSA' has a strikethrough, please edit.
Pg. 5 OSA a strikethrough has been corrected
Pg 5/6. I felt that Table 2 could be included as an Online Supplement and didn't need to be included in full in the paper.
Pg 5.6 We included Table 2 in the Supplement Data.
I loved the comparison of STOP-BANG to PSG, makes screening very accessible in a clinic setting. I also thought it very thorough to divide medication by type and look at the effect on variables of interest.
Although it is very attractive to examine separately the impact of every medication in the respiration of the patients, the number of patients is small to allow a such comparison. Thus we examined the different categories (N03, N05, N06) as a whole.
Pg 8. Please restructure Table 5 so that it is easier to read, put the values and labels on one line. It gets hard to read when decimal points drop off to different lines.
Pg. 8 We did it.
The conclusions are well thought out and helpful for both clinicians and researchers.